# Maternal Health and Nutrition Status, Human Milk Composition, and Growth and Development of Infants and Children: A Prospective Japanese Human Milk Study Protocol

**DOI:** 10.3390/ijerph17061869

**Published:** 2020-03-13

**Authors:** Keisuke Nojiri, Shunjiro Kobayashi, Satoshi Higurashi, Tomoki Takahashi, Yuta Tsujimori, Hiroshi M. Ueno, Shiomi Watanabe-Matsuhashi, Yasuhiro Toba, Junichi Yamamura, Taku Nakano, Kyoko Nomura, Toshiya Kobayashi

**Affiliations:** 1Research and Development Department, Bean Stalk Snow Co., Ltd., Saitama 350-1165, Japan; keisuke-nojiri@beanstalksnow.co.jp (K.N.); s-higurashi@beanstalksnow.co.jp (S.H.); tomoki-takahashi@beanstalksnow.co.jp (T.T.); yuta-tsujimori@beanstalksnow.co.jp (Y.T.); hiroshi-ueno@beanstalksnow.co.jp (H.M.U.); y-toba@beanstalksnow.co.jp (Y.T.); 2Megmilk Snow Brand Co., Ltd., Tokyo 160-8575, Japan; syunjirou-kobayashi@meg-snow.com (S.K.); s-matsuhashi@meg-snow.com (S.W.-M.); jyamamura@meg-snow.com (J.Y.); nakano@meg-snow.com (T.N.); 3Department of Environmental Health Science and Public Health, Akita University Graduate School of Medicine, Akita 010-8543, Japan; knomura@med.akita-u.ac.jp

**Keywords:** nutrition, human milk composition, development, growth, cohort study, Japanese, lactating woman, infant, child

## Abstract

It is unknown whether maternal health and nutrition are related to human milk composition or growth and development of infants and children. Here, we describe a protocol for a prospective five-year cohort study to clarify (i) how maternal health and nutrition, socioeconomic factors, and lifestyles affect human milk composition, and (ii) whether these are associated with growth and development of infants and children. In our study, we recruited 1210 Japanese mothers with singleton pregnancies from 73 obstetrics clinics and hospitals across Japan, between 2014 and 2019. We will measure the following: health information regarding maternal-child dyads using a self-administered questionnaire, maternal nutrition during breastfeeding using a Brief self-administrated Diet History Questionnaire, the development of infants and children using the Kinder Infant Development Scale, and the stress related to child rearing using the Mother’s Child Care Stress Scale. Simultaneously, we will collect human milk every 2 months during the first year after birth to measure its composition and levels of macronutrients. This study will generate useful data to investigate whether health status, nutritional status, lifestyle, and socioeconomic factors affect human milk composition and the growth and development of infants and children.

## 1. Introduction

An adequate supply of nutrition during “the first 1000 days of life”, i.e., the period from conception to 2 years of age, is essential to ensure the development and health of infants and children [1,2]. Human milk provides optimal nutrition for infants, as it contains various physiologically active substances in addition to fundamental nutrients such as proteins, fats, minerals, and vitamins [3]. In 2016, the Lancet published a special issue on breastfeeding that addressed its importance for both mother and child [4]. This overview revealed that the protective effects of breastfeeding have been investigated in relation to infectious morbidity and mortality, asthma, allergies, obesity, intelligence, and breast cancer. The effects of breastfeeding on growth and development in childhood have also been discussed [5,6].

Meanwhile, it has been reported that human milk composition is affected by ethnicity, environment, and lifestyle. For example, although human milk long-chain polyunsaturated fatty acids are important nutrients for infants, a meta-analysis showed that docosahexaenoic acid (DHA) concentrations in human milk were highly variable worldwide [7]. Another study emphasized the importance of ethnicity, reporting that human milk vitamin D levels varied by ethnicity [8]. We also found variance according to ethnicity in that high levels of osteopontin in breastmilk is more common in Asians than in Danes [9]. Hence, there is still room for consideration as to what factors affect the composition of human milk, and it is important to understand ethnicity, maternal diet, cultural background, and living environments before evaluating them. Furthermore, it remains unknown how potentially modified human milk composition, determined by maternal factors, may influence the growth and development of infants and children.

To date, several studies in Japan have investigated human milk. The study with the largest sample size, i.e., 100,000 mothers, is known as the Japan Environment and Child Study, which investigated environmental pollutants and chemical exposure affecting human milk during pregnancy [10]. The Hokkaido Birth Cohort Study focused mainly on chemical exposure and human milk [11]. The Kyushu Okinawa Maternal and Child Health Study was a prospective cohort study that investigated risk factors and preventive factors for maternal and child health problems such as allergic diseases, periodontal disease, postpartum depression symptoms, and childhood caries [12]. Although the last study [12] measured organochlorine compounds in human milk, none of these focused on the effects of maternal factors such as health, nutrition, and lifestyle on the growth and development of infants and children associated with human milk.

We are particularly interested in dietary habits, nutritional status, physical and psychological health status including maternal stress of child rearing, lifestyles, and the living environment of modern lactating Japanese women. In our study, we hypothesized that maternal factors affect breastfeeding practices and human milk compositions, which eventually determine the growth and development of infants and children (Figure 1). Therefore, we designed the study to investigate maternal health status, nutrition, lifestyle, and socioeconomic factors, and to clarify whether these factors are linked to human milk composition and/or are associated with the growth and development of infants and children. In this protocol paper, we describe the details of the study design and the future scope.

## 2. Materials and Methods

### 2.1. Study Design

This is a longitudinal prospective cohort study for Japanese lactating women and their infants and children. We will collect human milk and health information, and measure maternal dietary habits and nutritional status using a brief-type, self-administrated diet history questionnaire (BDHQ) [13]. We will measure the development of infants and children using the kinder infant development scale (KIDS) [14]. Details of each questionnaire are described in Section 2.4. Figure 2 outlines the study timeframe. Every 2 months during the first year after birth, we will collect human milk samples and obtain health information of mothers and infants, including BDHQ and KIDS data. After 1 year, follow-up on the participants will be performed. Health information and KIDS questionnaires will be completed at 18, 24, 36, 48, and 60 months. The collection of questionnaires and samples is currently ongoing.

### 2.2. Study Setting and Participants

Our study participants were recruited after delivery at 73 medical institutions comprising 16 hospitals and various obstetrics clinics from across all prefectures in Japan between October 2014 and May 2019 (enrolment period). We are following up individual participants for five years, and the study will be conducted between October 2019 and May 2024. Lactating women who visited hospitals and clinics were invited to participate in the study. At this time, they received adequate explanation regarding the study from our staff. Written informed consent was obtained from the respondents who agreed to participate in the study. Questionnaires and sterile plastic bags for collecting breast milk will be mailed to participants, and the completed questionnaires and breast milk will be returned to us. Participants who provide a valid questionnaire will receive a 1000 yen gift card, and those who provide breast milk will receive a 2000 yen gift card for each sample sent (1 U.S. dollar is approximately 110 yen). After that, we will conduct follow-up by mail, email, and phone for up to five years, at which time the study will be completed. Inclusion criteria were as follows: (1) healthy singleton infants aged <60 days old, and (2) healthy lactating women who had delivered infants and had not disrupted their breastfeeding patterns when providing human milk samples for this study. Exclusion criteria were as follows: (1) hepatitis B-positive or hepatitis C-positive participants, or participants with human immunodeficiency virus or human T-cell leukemia virus type 1 infections, (2) participants who are on medication for underlying illnesses, (3) participants who do not breastfeed, and (4) mothers, partners, or children who are not of Japanese ethnicity. Participants with the aforementioned illnesses were excluded because they were more likely to stop breastfeeding. We limited participation to Japanese ethnicity because we hypothesized that race is one of the fundamental determinants of breastfeeding practice. We recruited 1210 mothers; however, 83 did not provide valid responses, and 5 met the exclusion criteria. Finally, 1122 mothers became the subjects, who will be followed up on in the study.

### 2.3. Sample Size

Due to the scarcity of quantitative data on our hypothesis, we could not accurately calculate statistical power in this exploratory study. Consequently, we recruited 1210 mothers according to recruitment feasibility. To analyze correlations among the continuous variables of maternal nutrition, human milk composition, and infant/child growth and development, Pearson’s or Spearman’s correlation coefficient (r) will be used. The absolute value of r ranges from 0 to 1 with increasing magnitude indicating a stronger association. Given limited data on the aforementioned associations, a threshold of r = |0.2| was chosen to yield sufficient sample size. An r < |0.2| indicates a weak correlation that is considered too small to be clinically significant. We calculated 80% power and 5% type I error with a sample size of 194 [15]. According to a study in Japan, only 21% of lactating women achieved the WHO recommended 6 months of exclusive breast-feeding after birth [16,17]. In order to respond to this and evaluate the effects of breast-feeding while also considering the potential of participants dropping out of the study, approximately 1200 subjects are required for this study.

### 2.4. Data collection

#### 2.4.1. Questionnaire Survey for Health Information

Details of the variables assessed at each point are shown in Table 1. The questionnaire includes maternal information on socio-demographic factors (age, educational attainment, and employment status), physical and mental health (body weight, height, self-reported health conditions, child rearing-related stress, and a clinical history of underlying diseases and allergies), and lifestyle (eating habits, dietary supplement use, sleeping status, and smoking status). Information concerning pregnancy outcomes includes parity (nulliparity or multiparity), gestational age at delivery, and mode of delivery (cesarean section or otherwise). Information concerning infants and children includes sex, body weight, height, head circumference, age, feeding status including breastfeeding, clinical history of allergic and infectious diseases, immunizations, sleeping status, and bowel function status. Other family information includes family structure, annual household income, family history of allergies, partner’s educational attainment, employment status, and smoking status. As for the recording of health information such as the weight of the baby and changes in the weight of the mother before and after pregnancy, we asked mothers to refer to the maternal and child health handbook, which is a booklet to record information on maternal pregnancy, delivery, and child health conditions in Japan. Maternal child rearing-related stress will be assessed using a Mother’s Child Care Stress Scale, which consists of a checklist involving three subscales for mental and physical fatigue, anxiety over child rearing, and lack of husband’s support [18].

#### 2.4.2. Human Milk Sample Collection and Analyses

Human milk samples will be collected manually every 2 months up to 12 months postpartum. The mothers will be requested to collect milk after breastfeeding using a breast pump (Yanase Waichi, Japan). Principally, the mothers will collect 10–20 mL of human milk in a sterile plastic bag once a day for 7 d. Immediately following extraction, samples will be stored in a freezer at −18 °C prior to transfer via fast frozen delivery service to the laboratory and stored at −80 °C until thawing for analysis. Each frozen sample will be heated to 40 °C individually in a thermostatically controlled bath and pooled. Samples of 1–3 mL will then be homogenized using ultrasonication to measure energy, carbohydrate, protein, and lipid levels, and total solids in the milk using a validated human milk analyzer (Miris holding, Uppsala, Sweden) [19]. Trained researchers will measure all samples twice, and the averages of the two measurements will be used for analysis. Individual samples will be collected in plastic tubes and stored at −80 °C for further analyses. Components other than macronutrients can be measured depending on future hypotheses.

#### 2.4.3. Scale for Dietary Habits of Mothers

The dietary habits of the mothers will be assessed using the BDHQ [13], which was developed as a short version of the diet history questionnaire (DHQ) [20]. The DHQ yields information on the dietary intake of 150 food and beverage items and requires approximately 45–60 min to answer. Meanwhile, the BDHQ provides information on only 58 items, but requires only approximately 15–20 min to complete. The BDHQ is a four-page, fixed-portion questionnaire that assesses dietary intake based on the reported consumption frequency of 58 different food and beverage items. The BDHQ was found to be valid compared to dietary records in its ability to rank diets according to energy-adjusted intakes of many nutrients [21].

#### 2.4.4. Scale for the Development of Infants and Children

The scale for infant/child development will be scored according to KIDS, which was developed by the Center of Developmental Education and Research, Japan [14]. KIDS is a developmental screening scale that is convenient to use and easily completed by parents. It uses three different types of questionnaires (A, B, C). We will use type A for participants aged between 1 and 11 months, type B for participants aged between 12 and 24 months, and type C for participants aged between 36 and 60 months. Type A consists of a checklist involving six subscales for children’s behavior, namely physical/motor skills, manipulation, receptive language, expressive language, social relationships with adults, and feeding. Types B and C consist of nine subscales, with language concepts, social relationships with children, and discipline added to the 6 type a subscales. The checklist will be rated by a caregiver (typically a parent) with “yes” (i.e., 1) or “no” (i.e., 0) responses possible. Each subscale score will be added, with a higher score reflecting a higher level of development. Each subscale has options for ‘pass’ or a ‘fail’. Based on the answer sheet, the developmental quotient will be calculated as the sum of the scores on the nine subscales divided by the chronological age of the infant or child [22].

### 2.5. Data Analysis

Continuous variables will be expressed as mean ± standard deviation or as median with interquartile range, depending on the distribution. Categorical (nominal or ordinal) variables will be summarized as frequency and percentage. For statistical comparisons between a continuous variable and an ordinal variable, either a t-test or a Wilcoxon sum rank test will be used, depending on the distribution. For statistical comparisons between nominal variables, a chi-square test or a Fisher’s exact test will be performed. For continuous correlations, either Pearson’s or Spearman’s correlation coefficients are to be calculated. Multivariable models will be also applied including either logistic regression models or linear regression models for binary or continuous outcomes, respectively. In general, covariates adjusted in multivariable models will be selected at a significance level of *p* < 0.05 in univariate models. Other statistical approaches will be adopted in accordance with the nature of the factors and variables. All analyses are to be performed using R (version 3.6.1, R Foundation, Vienna, Austria) and SPSS Statistics 24 (IBM Inc., Chicago, IL, USA) software. A two-tailed p-value of ≤0.05 will be considered to be statistically significant.

### 2.6. Ethics

This study was undertaken in accordance with the Ethical Guidelines for Clinical Research (Ministry of Health, Labor, and Welfare, Japan) recommendations and was approved by the Ethics Committee of the Fukuda Internal Medicine Clinic (IRB No. 20140621-03). All participants provided written informed consent in accordance with the Declaration of Helsinki. This study was registered with the University Hospital Medical Information Network Clinical Trials Registry (UMIN ID 000015494, 2014).

## 3. Discussion

This prospective cohort study will investigate healthy lactating women who were recruited at obstetric clinics across Japan. We will collect various health information on maternal-child dyads with a sample size of over 1000. These will include the physical and psychological status of mothers, maternal nutrition, dietary patterns, lifestyles, and living environments of lactating women. We will also collect human milk samples every two months during the first year after birth. The prospective study design will allow us to verify the hypothesis that maternal health and nutrition affect human milk composition, and ultimately, the growth and development of infants and children.

In this study, we experienced high response rates (1122/1210, 93%) to the first survey completed at 1–2 months postpartum. Such a high response rate may make results generalizable to some extent, but requires careful attention, in that our participants are healthier mothers compared to mothers who gave birth at hospitals in Japan. With monetary incentives, we hope to have a high follow-up rate in the prospective cohort study yielding high-quality evidence.

We used valid and reliable scales for maternal nutrient consumption and infant/child development. The BDHQ, a brief version of DHQ, requires only 10–15 min to complete and is valid and reliable for the evaluation of nutritional habits [21]. In this study, we will measure the BDHQ six times in total, which will enable us to investigate seasonal variety and obtain more accurate values for unstable nutrients, such as retinol and n-3 polyunsaturated fatty acids [21]. We have adopted KIDS to evaluate infant/child development, which was developed in 1989. This KIDS was standardized using 6000 Japanese infants and children aged 0 to 6 years old, recruited from 38 prefectures [14]. The advantages of KIDS include the ease of assessment based on self-reporting by mothers, indicating that children do not have to see medical doctors. KIDS also evaluates nine aspects of infant/child development: physical/motor skills, manipulation, receptive language, expressive language, social relationships with adults, feeding, language concepts, social relationships with children, and discipline. Unlike previous studies that evaluated few aspects of development [23,24,25], this multifaceted evaluation extends the analytical approach to investigate relationships between infant/child development, maternal nutrition, and human milk composition.

Human milk constitutes a comprehensive source of macro- and micro-nutrients and bioactive factors, which are essential for the growth and development of infants and children [3]. The amount and composition of expressed human milk varies depending on the individual, and can vary even in the same mother. The composition of human milk changes with the nutritional status of mothers, including consumption of DHA and vitamins [26,27]. However, the relationship between maternal nutrition and human milk is not consistently reported. For example, a study of 74 human milk samples reported that vitamin B-12 in human milk is independent of the maternal diet pattern [28]. Another study suggested that the nutritional and immune components of human milk are in fact influenced by maternal psychological status [29]. The milk components evaluated in these studies include oligosaccharides, bioactive proteins such as antibodies and cytokines, and nonprotein nitrogen, such as nucleic acid compounds. Hence, how maternal health and nutrition determines human milk composition requires further accumulation of scientific findings. A longitudinal study design and sufficient amounts of human milk samples will minimize measurement variabilities, and thus, allow for the accurate investigation of relationships between maternal health and nutrition, human milk composition, and growth and development of infants and children.

There are several limitations of the study described herein. First, the human milk will be voluntarily collected after breastfeeding, which may introduce a random error in timing. It has previously been shown that human milk composition can change according to the timing of excretion (i.e., fore, middle, and hind milk phase) [30]. Previous studies collected middle or hind milk, but we will not specify the exact timing of individual milk sample collection. We will ask mothers to collect human milk samples after breastfeeding so that milk sampling will not interfere with maternal breastfeeding. This suggests that stress should be reduced in mothers, which may increase the follow-up rate in such studies. Second, our participants are likely to be healthier than the general population because of recruitment of almost all participants in the clinic. Caution will be required when extrapolating results to other populations. Third, health information, BDHQ, and KIDS will be measured based on self-reporting, which could cause biases, and thus, will require careful interpretation.

## 4. Conclusions

In this longitudinal prospective study, we will collect various data on Japanese lactating women and their children for 5 years and evaluate human milk periodically during the first year after birth. Such studies can clarify how maternal health and nutritional status, lifestyle, and socioeconomic factors affect breastfeeding practice and human milk composition, which will eventually determine the growth and development of infants and children.

## Figures and Tables

**Figure 1 ijerph-17-01869-f001:**
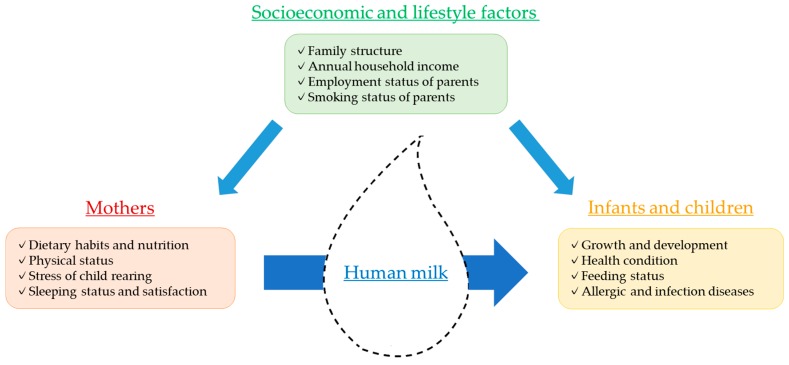
Relationship between mothers, human milk, infants/children, and socioeconomic and lifestyle factors.

**Figure 2 ijerph-17-01869-f002:**
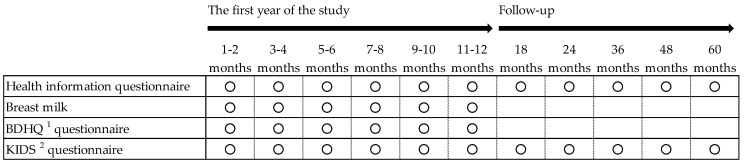
The study timeframe. ^1^ BDHQ, brief-type self-administrated diet history questionnaire. ^2^ KIDS, kinder infant development scale.

**Table 1 ijerph-17-01869-t001:** Summary of data collection for health information questionnaire.

Variables	Pre-Pregnancy	During Pregnancy	After Delivery (month)										
			1–2	3–4	5–6	7–8	9–10	11–12	18	24	36	48	60
Mother													
Age			X										
Body weight	X	X	X	X	X	X	X	X	X	X	X	X	X
Height			X										
Health condition		X	X	X	X	X	X	X					
General and childcare-related stress			X	X	X	X	X	X	X	X	X	X	X
History of underlying disease and allergies			X								X	X	X
Dietary supplement use		X	X	X	X	X	X	X					
Sleeping status			X	X	X	X	X	X					
Smoking status		X	X	X	X	X	X	X	X	X	X	X	X
Parity (nulliparity or multiparity)		X											
Gestational age at delivery		X											
Mode of delivery		X											
Educational attainment			X										
Employment status			X						X	X	X	X	X
Infant and child													
Sex			X										
Birth body weight, height and head circumference			X										
Body weight, height, head circumference			X	X	X	X	X	X	X	X	X	X	X
Days postpartum			X	X	X	X	X	X					
Feeding status			X	X	X	X	X	X	X	X	X	X	X
Development of allergic and infectious diseases			X	X	X	X	X	X	X	X	X	X	X
Immunizations status			X	X	X	X	X	X	X	X	X	X	X
Sleeping status			X	X	X	X	X	X			X	X	X
Bowel function status.			X	X	X	X	X	X					
Family members													
Family structure			X						X	X	X	X	X
Annual household income			X						X	X	X	X	X
Clinical history of allergies			X								X	X	X
Educational attainment of partner			X										
Employment status of partner			X						X	X	X	X	X
Smoking status of partner			X	X	X	X	X	X	X	X	X	X	X

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
