# Peer review of "Maternal Health and Nutrition Status, Human Milk Composition, and Growth and Development of Infants and Children: A Prospective Japanese Human Milk Study Protocol"

_ijerph, 2020, doi:10.3390/ijerph17061869_

Round 1
Reviewer 1 Report
The study is a protocol paper describing a longitudinal follow up looking at if health status, nutritional status, and other lifestyle and socioeconomic factors affect human milk composition and the growth of offspring.
-Recent literature talks about the effect of maternal alcohol and marijuana use. May want to include those variables if including smoking.
-Highly recommend to ask maternal educational status
-Highly recommend to ask at least infant weight/height as well as maternal weight if not measuring them directly at each visit
-There is no information or section talking about how to approach and recruit/screen women, who and how to consent women, the setting to collect information and data, length of each visit, compensation associated with each visit, and who is conducting these followups.
-If you need only 223 women, why did you recruit 1122 mothers?
-In the study design section, you may not want to talk about specific types of questionnaires too detailed and confusing if you are talking more details later on.
-In data analysis, there is nothing mentioned about the criteria to include which variables as covariates in multivariate analyses.
Reviewer 2 Report
The revised manuscript looks better and the authors have addressed my question.
Reviewer 3 Report
Authors have addressed most of my comments, and the manurscipt has been improved much. The following issue should be addressed further.
- Table 1, it was not clear, because it was not presented clearly about the information about pre-regnancy, and it seemed to mix with informaiton about pregnancy.
- I am supprised that authors deleted some results from this cohort study, but I think they are necessary if posssible.
Round 2
Reviewer 1 Report
Please also clarify whether participants received any monetary or non monetary compensation for participation and how long each follow up took by email, phone, and in person.
Author Response
Please see the attachment.

This manuscript is a resubmission of an earlier submission. The following is a list of the peer review reports and author responses from that submission.
Round 1
Reviewer 1 Report
The study is a nationwide survey on maternal nutritional status using validated instruments following up to 60 mos among breastfeeding mothers in Japan. The survey itself is significant however the findings are nowhere at the stage of presenting and publishing esp with no comparisons for Table 3 and at such an early stage of data collection.
It was not clear which points are the strength and limitation in Discussion.
Eligibility criteria do not seem to match with the survey timepoints. Not sure of the reason why Hep B and C positive mothers were excluded.
Reviewer 2 Report
In this study “Relationship between maternal nutrition, breast milk composition, and the growth and development of infants and children: protocol and baseline characteristics for the Japanese Human Milk Study” authors discussed their protocol and baseline characteristics. While this is a very important topic, in this submitted study, analysis of this baseline information does not provide much information. I will be curious to find out the results of the prospective study.
Reviewer 3 Report
This is profile of study in Japan women and children. It wanted to investigate the effect of a mother's nutritional status on breast milk composition and whether it influenced infant and childhood growth. At present, they reported the profile of study design and baseline information on mother and child. However, the following technical problems in methods should be addressed further in order to improve the manuscript.
Authors should add some statement on maternal nutrition in Japan and their possible problems. About study design, clear statement should be provided, so some changes should be done about each part. This is a cohort study, so authors should provide clearly the duration of this study; the first participant entered the cohort in 2014, the las one in 2019, right? For each pairs of mother and child, totally how long will they be followed. That is to say, the duration of study should be clearly reported. The lactating women were included? Table 1 presented the information collected in the study, but the time to collect each kind ofinformation remains unclear in text. For example. Maternal and infant questionnaire will be conducted six times? If yes, for each time they are different completely? In text, this questionnaire of mother just collect baseline information, why did they not collect the maternal weight during pregnancy but body weight before pregnancy? Maternal BDHQ will be collected every two month? It seems high frequent. Why did they do like that? In methods, authors should present the main variable of interest or their logical ideal for this study because this was a profile. In the results, they reported the baseline information of mother and child, but they seemed not to report the breast milk composition, as what authors said, this study began in 2014. Maybe there were some results about breast milk. They emphasized frequently the key outcome was breast milk composition. About sample size, authors estimated it by using prevalence rate for termination of breastfeeding. Why did you do it? In this study, main outcome is maternal nutrition, milk composition, and child growth. So estimation of sample size should focus on these indicators based on effect of maternal nutrition on milk composition, and child growth. If they want to focus on milk composition, this indicator should be used for estimation of sample size. Due to multiple outcomes, authors should provide strategy about sample size. Authors should provide some statement on possible future plan by using data from this cohort in the section of discussion.